# Reimagining Digital Gazetteers : A Wikidata-Powered Approach

## Maxime Guénette

**Keywords**: gazetteers, Linked Open Data, Wikidata, digital cultural heritage

As an open, multilingual, and collaborative knowledge base, Wikidata is increasingly essential for academic research, particularly in Digital Humanities (DH). Its capacity to centralize data from multiple sources, structure it using interoperable standards, and enrich it through collaboration makes it invaluable for DH projects that both import and export data directly on the platform.

A prevalent type of project in DH is the development of digital gazetteers. A gazetteer is traditionally a directory of location names and coordinates. However, in its digital form, it links locations to enriched data such as historical descriptions, spatial coordinates, and temporal information. Digital gazetteers have increase in popularity since the early 2000s for their ability to publish geographic data using Semantic Web standards. Nevertheless, numerous reports from the scientific community indicate that the publication of Linked Open Data (LOD) through digital gazetteers remains hindered by several barriers, including high technical skill requirements and significant financial costs.

This paper demonstrates Wikidata's potential for creating state-of-the-art digital gazetteers through two case studies from classical studies and archaeology. These examples illustrate how Wikidata supports both micro- and macro-scale gazetteer projects, enabling advanced data integration, spatial analysis, and collaboration.

The first case study focuses on the International (Digital) Dura-Europos Archive (IDEA) project, which uses Wikidata to build an urban gazetteer of Dura-Europos, an ancient city in Syria. The city's cultural heritage is under threat due to the ongoing civil war. By leveraging Wikidata's multilingual capabilities and Linked Open Data principles, IDEA aims to reassemble fragmented data from Dura-Europos located in collections worldwide. This effort addresses historical and archival biases from colonial-era excavations, promoting more equitable access to heritage. Wikidata's collaborative nature enables for the first time Syrian researchers and the public to contribute to and benefit from the project.

The second case study examines our doctoral research on sacred spaces in Roman Britain. As part of the Wikiproject *Temples in Roman Britain*, we are cataloging temples and sanctuaries in the Roman province of Britannia (43–410 AD), with metadata such as construction and destruction dates, geographic coordinates, connections to other gazetteers, and interpretative frameworks. Furthermore, the project uses the SPARQLing Unicorn plugin for QGIS, enabling dynamic integration of GeoJSON layers directly from Wikidata's LOD ecosystem, facilitating spatial analysis and visualization.

Both projects follow similar methodologies, using legacy data to transform disparate archival records into structured and interoperable datasets. It includes extracting information into spreadsheets, modeling the data to fit Wikidata's ontology, and preparing it for upload using OpenRefine. Schemas are then used to ensure consistency, and the data is exported through QuickStatements for quality control before batch uploads to Wikidata. This workflow ensures data accuracy and integration into the Linked Open Data ecosystem.

By highlighting these case studies, this paper argues that Wikidata is not only a reliable platform for digital gazetteers but also a transformative tool for DH. Its ability to democratize data creation, integrate Semantic Web technologies, and foster global collaboration represents a significant advancement in the creation and publication of linked geographic data.

