# OpenReview forum: "Reimagining Digital Gazetteers: A Wikidata-Powered Approach"
_wikimedia.it/Wikidata_and_Research/2025/Conference — WD&R Paper_

### Official Review · ~Annick_Farina1 · 2025-01-09
**interesting analisis of Wikidata potential for digital gazetteers and as a transformative tool for DH**

**Originality:** 5
**Impact:** 4
**Confidence:** 3

**Review:**

The authors would like to illustrate two case studies a) the International (Digital) Dura-Europos Archive (IDEA) project, which uses Wikidata to build an urban gazetteer of Dura-Europos, an ancient city in Syria and b) their doctoral research on sacred spaces in Roman Britain as part of the Wikiproject Temples in Roman Britain (they cataloged temples and sanctuaries in the Roman province of Britannia). From these case of studies the authors would show how Wikidata is not only a reliable platform for digital gazetteers but also a transformative tool for DH.

**Compliance:**

5

**Scientific Quality:**

4

---

### Official Review · ~Monica_Berti1 · 2025-01-10
**A very interesting paper on the use of Wikidata for academic research in historical geography and digital gazetteers**

**Originality:** 5
**Impact:** 5
**Confidence:** 5

**Review:**

The author of this paper presents the use of Wikidata for the development of digital historical gazetteers. This paper demonstrates the potential of Wikidata for the creation of state-of-the-art digital gazetteers through two case studies from Classical Studies and Archaeology: data from 1) the International (Digital) Dura-Europos Archive (IDEA) project and from 2) a doctoral dissertation on sacred spaces in Roman Britain. This paper shows that the author is familiar with the use of Wikidata and DH formats and standards. He also presents concrete challenges and needs of the community for the future of digital historical geography.

**Compliance:**

5

**Final Paper Review:**

I confirm my original review of the abstract also for the paper, which demonstrates the potential of Wikidata for the creation of state-of-the-art digital gazetteers through two case studies from Classical Studies and Archaeology: 1) the doctoral dissertation of the author on sacred spaces in Roman Britain and 2) the International (Digital) Dura-Europos Archive (IDEA) project. The paper shows that the author is familiar with the use of Wikidata and DH formats and standards. He also presents concrete challenges and needs of the community for the future of digital historical geography.

**Scientific Quality:**

5

---

### Decision · Program_Chairs · 2025-02-05

Accept (Paper)